# Structural basis of Nrd1–Nab3 heterodimerization

Belén Chaves-Arquero[1,5] ⓘ, Santiago Martínez-Lumbreras[1,6] ⓘ, Sergio Camero[1] ⓘ, Clara M Santiveri[2] ⓘ, Yasmina Mirassou[1,3] ⓘ, Ramón Campos-Olivas[2] ⓘ, Maria Ángeles Jiménez[1] ⓘ, Olga Calvo[4] ⓘ, José Manuel Pérez-Cañadillas[1] ⓘ

Heterodimerization of RNA binding proteins Nrd1 and Nab3 is essential to communicate the RNA recognition in the nascent transcript with the Nrd1 recognition of the $Ser_5$-phosphorylated Rbp1 C-terminal domain in RNA polymerase II. The structure of a Nrd1–Nab3 chimera reveals the basis of heterodimerization, filling a missing gap in knowledge of this system. The free form of the Nrd1 interaction domain of Nab3 (NRID) forms a multi-state three-helix bundle that is clamped in a single conformation upon complex formation with the Nab3 interaction domain of Nrd1 (NAID). The latter domain forms two long helices that wrap around NRID, resulting in an extensive protein–protein interface that would explain the highly favorable free energy of heterodimerization. Mutagenesis of some conserved hydrophobic residues involved in the heterodimerization leads to temperature-sensitive phenotypes, revealing the importance of this interaction in yeast cell fitness. The Nrd1–Nab3 structure resembles the previously reported Rna14/Rna15 heterodimer structure, which is part of the poly(A)-dependent termination pathway, suggesting that both machineries use similar structural solutions despite they share little sequence homology and are potentially evolutionary divergent.

## Introduction

The mechanisms of transcription termination have been profusely studied from different approaches; from cell biology to structural methods (Richardson, 1996; Birse et al, 1998; Dichtl & Keller, 2001; Mischo & Proudfoot, 2013; Arndt & Reines, 2015; Lemay & Bachand, 2015; Porrua & Libri, 2015). In the *Saccharomyces cerevisiae* model system there, are two different transcription termination mechanisms: the poly(A)-dependent pathway that mainly processes

mRNAs (Birse et al, 1998; Dichtl & Keller, 2001) and the poly(A)-independent pathway that processes most of the short noncoding transcripts such as snoRNAs (Conrad et al, 2000; Carroll et al, 2004, 2007; Kim et al, 2006). This latter pathway principally involves three proteins, Nrd1, Nab3, and Sen1, and is referred to as the Nrd1-Nab3-Sen1 (NNS) pathway. Interestingly, in both pathways, the biochemical activities are performed by protein machineries rather than by ribonucleoprotein assemblies, as in RNA splicing. Although the two pathways involve specific proteins, the two transcription termination routes use remarkably similar strategies to precisely identify the termination signal. First, the two pathways include proteins with interaction domains (CID) capable to recognise the C-terminal domain (CTD) of the Rpb1 subunit of RNA Pol II: Pcf11 in the poly(A)-dependent pathway and Nrd1 in the NNS one. The CTD contains heptapeptide repeats with the consensus sequence YSPTSPS (Allison et al, 1985; Corden et al, 1985), tightly regulated by post-translational modifications such as phosphorylation of serines 2, 5, and 7 (Hirose & Manley, 2000; Hsin & Manley, 2012; Zaborowska et al, 2016; González-Jiménez et al, 2021). Different CIDs have different specificity; for instance, Pcf11 interacts with $CTD-Ser_2-P$ (Meinhart & Cramer, 2004), whereas Nrd1 recognises $CTD-Ser_5P$ (Vasiljeva et al, 2008; Kubicek et al, 2012). Because the CTD phosphorylation pattern changes along transcription, the specific interaction with Nrd1 or Pcf11 allows a differential timing in the recruitment of each associated machinery: the NNS complex is recruited early during transcription and poly(A)-dependent complexes much later (Mischo & Proudfoot, 2013; Porrua & Libri, 2015). In addition, the Nrd1 CID can recognise other peptides from Trf4 (Tudek et al, 2014) and Sen1 (Zhang et al, 2019; Han et al, 2020) and plays an important role in coordinating different steps along the pathway. Such promiscuity has not been reported for the Pcf11-CID, but it would not be surprising that it could recognise peptides different from the CTD. The second resemblance between poly(A)-dependent and NNS pathways is the presence of RNA binding proteins (RBPs) with different degree of sequence specificity. In the first pathway, Hrp1 and Rna15

[1]Departamento de Química-Física Biológica, Instituto de Química-Física "Rocasolano" (IQFR), Consejo Superior de Investigaciones Científicas (CSIC), Madrid, Spain [2]Spectroscopy and Nuclear Magnetic Resonance Unit, Structural Biology Programme, Spanish National Cancer Research Centre, Madrid, Spain [3]Centro Nacional de Análisis Genómico (CNAG)-CRG, Centre for Genomic Regulation (CRG), The Barcelona Institute of Science and Technology, Barcelona, Spain [4]Instituto de Biología Funcional y Genómica, Consejo Superior de Investigaciones Científicas, Universidad de Salamanca, Salamanca, Spain [5]Research Department of Structural and Molecular Biology, University College London, London, UK [6]Institute of Structural Biology, Helmholtz Zentrum München, Neuherberg, Germany and Bavarian NMR Centre, Chemistry Department, Technical University of Munich, Garching, Germany.

Correspondence: jmperez@iqfr.csic.es
Belén Chaves-Arquero's present address is Centro de Investigaciones Biológicas "Margarita Salas" (CIB), Consejo Superior de Investigaciones Científicas (CSIC), Madrid, Spain.

recognise a specific termination signal in the 3′-UTR via RNA recognition motif (RRM) domains: two on Hrp1 that interact with the polyadenylation enhancement element (Pérez-Cañadillas, 2006) and one in Rna15 that binds U-rich sequences (Pancevac et al, 2010). Furthermore, both RBPs act co-ordinately to recognise longer RNA segments (Leeper et al, 2010). In the NNS pathway, two RBPs, Nab3 and Nrd1, likewise contain RRM domains that contact specific termination signals (Hobor et al, 2011; Lunde et al, 2011; Franco-Echevarría et al, 2017). In the case of Nrd1, the unusual structure of its RNA-binding domain (RBD) allows specific interactions with relatively short RNA terminators (Franco-Echevarría et al, 2017). Therefore, the main RNA recognition activity in both pathways relies on two pairs of RBPs (Hrp1/Rna15 and Nab3/Nrd1) and occasionally in other proteins like Sen1, in the NNS route, that binds nascent RNA with less specificity. However, the functions of the RBPs are not limited to RNA recognition: they are also involved in protein–protein interactions—the third similarity between both pathways. For instance, in the poly(A)-dependent pathway, the hinge domain of Rna15 interacts with the Rna14 Monkeytail domain (Moreno-Morcillo et al, 2011). Moreover, Rna14 interacts with Hrp1 via their HAT repeats, using an interaction surface compatible with RNA binding (Barnwal et al, 2012). In the NNS pathway, Nrd1 and Nab3 coordinate their RNA-binding activities by heterodimerization (Conrad et al, 2000; Vasiljeva et al, 2008). Although the regions involved in this interaction have been known for a long time, the structural bases of the heterodimer formation remain elusive.

Here we characterize the structural propensities of the Nrd1 and Nab3 heteromerization domains in their free states along with their interaction using a combination of nuclear magnetic resonance (NMR), circular dichroism (CD), and isothermal titration calorimetry (ITC) techniques. More importantly, we unveil the structural basis of Nrd1–Nab3 heterodimerization by solving the NMR structure of a chimeric construct that includes regions of the two proteins in a single polypeptide, which is a bona fide model of the actual heterodimer. Based on this high-resolution structure we identify key residues at the Nrd1–Nab3 interface and study the effect of their mutation in vivo, unveiling their physiological impact in yeast fitness. Finally, the Nrd1–Nab3 chimera displays significant resemblance to the Rna14/Rna15 heterodimer, suggesting that both transcription termination pathways share similar strategies to recognise RNA terminators.

## Results

### Isolated Nrd1 and Nab3 heterodimerization domains show different levels of structure

The Nrd1 interaction domain of Nab3 (Nab3$_{191-261}$) (NRID) and the Nab3 interaction domain of Nrd1 (Nrd1$_{147-222}$) (NAID) are the two regions involved in Nrd1–Nab3 heterodimerization (Conrad et al, 2000; Vasiljeva et al, 2008) (Fig 1A). The fragments, of about 70–80 residues in length, show high conservation of both hydrophobic and polar amino acids (Figs 1A and S1A and B), suggesting that heterodimerization may be accomplished by a

combination of polar and nonpolar contacts. We started the study by analyzing the structural properties of these two domains in isolation.

Nrd1$_{147-222}$ is located between the CID and RNA recognition domains (Steinmetz & Brow, 1996; Conrad et al, 2000) (Fig 1A). At first sight its $^{1}$H-$^{15}$N HSQC spectrum is typical of an intrinsically disordered protein: the amide cross-peaks are sharp and poorly dispersed (Fig 1B, left panel). However, the number of signals is lower than expected and the assignment process confirmed the lack of backbone amide cross-peaks for large regions of the construct (residues marked with a star in Fig 1C left panel). The secondary structure propensities for the observable residues, as obtained by $^{1}$H/$^{13}$C conformational chemical shifts, show that they are predominantly unstructured (grey bars in Fig 1C, left panel). The missing cross-peaks could be explained by conformational exchange broadening and/or participation of those regions in high molecular weight oligomerization, whose broad NMR line widths are beyond detection, leaving the flexible tails with faster dynamics as the only "visible" parts in NMR. In addition, these putative interactions might be heterogeneous, resulting in a further NMR signal broadening through conformational exchange processes. The CD spectrum of Nrd1 NAID (Fig 1D in blue) reveals a mixture of unstructured and α-helical conformation, which points to the α-helical nature of these hypothetical oligomers.

On the other hand, Nab3 NRID (residues 191–261), placed between an acidic region of unknown function and the RNA binding domain (Wilson et al, 1994; Conrad et al, 2000) (Fig 1A), shows a $^{1}$H-$^{15}$N HSQC with greater signal dispersion than Nrd1 NAID evidencing some residual structure (Fig 1B, right panel). The $^{1}$H/$^{13}$C conformational chemical shifts allow us to identify two stable α-helices spanning residues 209–219 and 235–244 (Fig 1C, right panel). This was corroborated by the CD spectrum showing the two characteristic α-helix minima (Fig 1D in red). The relative intensities of the 208 and 222 nm minima are inverted relative to the typical CD spectrum of α-helix and this is a feature observed in helical bundles or coiled-coil CD spectra (Greenfield, 2006). To get a more accurate picture of the Nab3 conformation, we determined the 3D NMR structure of an Nab3 NRID$_{198-250}$ construct, devoid of flexible N- and C-terminal flanking parts. The 2D NOESY spectrum of this domain is dominated by short and medium-range nuclear overhauser effects (NOEs) characteristic of helical structures, but long-range cross-peaks between Phe 229 and Ile 241 and between Val 215 and Ile 241 can also be observed (Fig S2). The final NMR structure of Nab3 heterodimerization domain has a well-defined secondary structure in the helical regions, but an ill-defined tertiary fold (Fig 1E). These structured regions comprise a long N-terminal α-helix (208–221) and a short C-terminal α-helix (239–246) that interacts with the inter-helical linker (Ile 241-Phe 229) (Fig 1E). This interaction nucleates a minimal hydrophobic core, which is not large enough to stabilize the protein in a single conformation. It is likely that such internal flexibility of the molecule affects line widths of the $^{1}$H-$^{15}$N HSQC signals, making them broader than expected for a molecule of its size.

In summary, Nrd1 and Nab3 heterodimerization domains have different structural behavior in isolation. Nab3 NRID shows higher α-helical content and forms a loose association of two helices, whereas Nrd1 NAID is less structured and with a large region

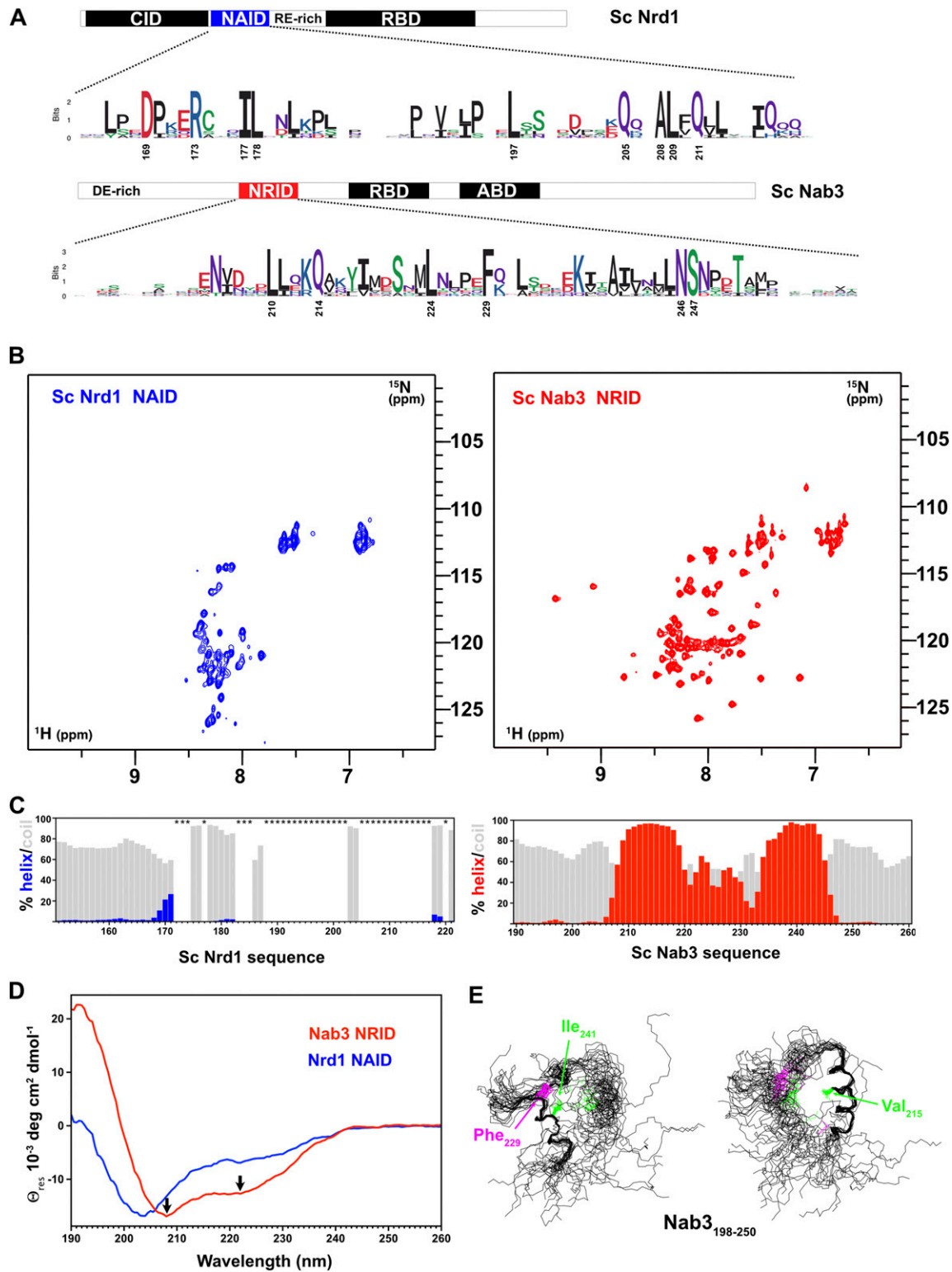

**Figure 1. Structural data for the isolated Nrd1 and Nab3 heterodimerization domains.**
**(A)** Schematic representation of Nrd1 and Nab3 domain architecture with the heterodimerization domains NAID and NRID highlighted in blue and red, respectively. Sequence logos to represent the amino acid conservation of these domains have been produced from sequence alignments of Nrd1 and Nab3 orthologs of organisms of the *Saccharomyces* clade (full sequence alignments in Fig S1). Other domains/regions are displayed: CID (CTD interaction domain), RBD (RNA binding domain), ABD (tRNA anticodon binding domain), DE-rich (acidic region), and RE-rich (arginine/glutamic-rich region). **(B)** $^{1}$H-$^{15}$N HSQC spectra of Nrd1 NAID (residues 147–222) (left panel in blue) and Nab3 NRID (residues 191–261) (right panel in red) in their isolated forms. **(C)** Percentage of secondary structure calculated from $^{13}$C/$^{1}$H chemical shifts for Nrd1 NAID (left panel) and Nab3 NRID (right panel). The bar charts indicate the percentage of α-helix (blue/red bars) versus random coil (grey bars) calculated with the program

involved in conformational heterogeneity and/or multimerization processes.

**Nrd1–Nab3 heterodimerization**

Next, we monitored the formation of the Nrd1–Nab3 heterodimer by NMR. Titration of unlabelled $Nab3_{191-261}$ on $^{15}N$-labelled $Nrd1_{147-222}$ prompts dramatic changes in the $^{1}H$-$^{15}N$ HSQC spectrum, with new signals appearing and most becoming disperse because of the induction of structure (blue versus grey signals in Fig 2A, left panel). Now all the expected NMR signals are observed, in contrast with the free state (grey signals in Fig 2A) showing that Nrd1 NAID adopts a single and unique conformation upon binding to Nab3. The $^{1}H/^{13}C$ conformational shifts of the bound state reveal that the adopted structure includes two long helices spanning residues 170–179 and 202–219 (Fig 2B, left panel). Remarkably, these helices correspond to the regions with missing cross-peaks in the free state (Fig 1C, left panel). On the other hand, titration of unlabelled $Nrd1_{147-222}$ over $^{15}N$-labelled $Nab3_{191-261}$ also causes large changes on the $^{1}H$-$^{15}N$ HQSC spectrum compared with that of the free form (red versus grey signals in Fig 2A, right panel). However, the secondary structure profile remains almost identical (Fig 2B, right panel) to that of the free state (Fig 1C, right panel), suggesting that the secondary structure elements are preconfigured in Nab3 free state.

We also analyzed the binding energetics of this protein–protein interaction by ITC using two Nrd1 constructs: $Nrd1_{1-222}$, including the CID (Vasiljeva et al, 2008), and $Nrd1_{147-222/290-489}$ also encompassing the RBD (Franco-Echevarría et al, 2017) but lacking residues 223–289 (Fig 2C). Those contain the RE-rich region (Fig 1A) and were removed because the recombinant proteins including them expressed as insoluble proteins. The interaction energies are very similar for both Nrd1 constructs, with $K_D$ in the nanomolar range. The dissociation constant of the $Nrd1_{1-222}/Nab3_{191-261}$ complex is almost identical to the previously reported 160 nM value for $Nrd1_{6-224}/Nab3_{204-248}$ (Vasiljeva et al, 2008). However, the stoichiometry is lower in our experiments (0.4 versus 1.0). In contrast, the $Nrd1_{147-222,290-489}$ complex shows a stoichiometry closer to one and a ~fourfold tighter binding (Fig 2C). These values are reproducible (Fig S3A) and suggest that the CID might have some destabilizing effect on the heterodimer. To corroborate that this effect is specific to the CID, we performed the ITC experiments with a construct replacing the CID by an unrelated tag of similar size (*Escherichia coli* TxA). The resulting $K_D$ values were tighter and comparable to that of the $Nrd1_{147-222,290-489}$ interaction (Figs 2C and S3B), further backing the slight destabilizing effect of the CID on heterodimerization. Surprisingly, we obtained stoichiometries below 1 in both cases, but the formation of an Nrd1/Nab3 2:1 heterodimer (that would result in N = 0.5) has not been reported despite the large amount of data available for this system. Instead, a simpler explanation for this behavior would be that part of Nrd1 forms kinetically trapped aggregates that reduce its effective concentration capable to interact with Nab3.

**An Nrd1–Nab3 chimera reveals the key structural elements of heterodimerization**

Progress in the structural understanding of the Nrd1–Nab3 requires a more accurate model than the previous approaches. However, the structural determination of the Nrd1–Nab3 heterodimer by NMR faces the challenge of preparing a highly homogeneous complex. The heterogeneity of $Nrd1_{147-222}$, particularly the likely presence of kinetically trapped aggregates, makes impossible to obtain data of enough quality for the structure determination. Regular and isotope-filtered NOESY spectra were poor and suffered from chemical exchange effects and spurious cross-peaks that degrade their quality. Therefore, as an alternative to overcome these technical difficulties, we constructed Nrd1–Nab3 chimeras.

In a first design, we concatenated conserved regions of Nrd1 and Nab3 (chimera $Nrd1_{147-222}$-$Nab3_{202-261}$). Most of the signals observed in the $^{1}H$-$^{15}N$ HQSC spectrum of this chimera are equivalent to cross-peaks present in the sub-spectra of Nrd1/Nab3 in their bound forms (Fig S4A). Indeed, the chemical shift differences are only noticeable for the first residues of Nab3 in the chimera, just after the connection point between the two proteins (Fig S4B). To optimize the design, we trimmed the flexible residues at both ends (characterized by high intensity peaks in the HSQC and not heterodimer-induced secondary structure; Fig 2B), and added a 16-residue flexible linker. This version, $Nrd1_{168-220}$-GGGSGGGTGGGTGGGS-$Nab3_{203-254}$, and the next one, $Nrd1_{168-222}$-$Nab3_{202-261}$, have larger chemical shifts differences with the heterodimer sub-spectra (Fig S4B), and, most importantly, a sub-set of minor signals appeared; indicative of the presence of minor forms. Moreover, these versions are less stable (their HSQC change within 1–2 d). The addition of 10 more residues at the N terminus, solves the heterogeneity and stability problems. This construct, $Nrd1_{158-222}$-$Nab3_{202-261}$, gives an excellent NMR spectrum (Fig 3A), and shows nearly identical chemical shift differences than the first construct ($Nrd1_{147-222}$-$Nab3_{202-261}$). This validates this chimeric construct as a faithful model of the heterodimer, and consequently, we proceeded to determine its 3D structure. The low abundance of aromatic residues and the large proportion of methyl-containing amino acids cause a high overlap in the methyl region of the $^{1}H$-$^{13}C$ HSQC that can be alleviated with $^{13}CH_3$-specific labelling (Fig S5B) using α-ketoacid precursors in combination with $^{13}C$-edited 3D NOESY experiments (Fig S5B). As a result, the final structure calculation used a large number of distance restraints allowing to obtain a highly accurate model (Fig 3B).

The structure of the Nrd1–Nab3 chimera presents an unusual α-helical arrangement that reveals the structural basis of Nab3/

---

d2D+ (Camilloni et al, 2012). Other types of secondary structures have been omitted because of their low calculated percentages. Nrd1 NAID residues with missing HQSC cross-peaks are indicated with stars. **(D)** Superposition of the circular dichroism spectra of $Nrd1_{147-222}$ (in blue) and $Nab3_{191-261}$ (in red). Black arrows mark the position of the two typical minima at 208 and 222 nm exhibited by α-helix structures. **(E)** Superpositions of the 20 lowest target function conformers calculated for Nab3 NRID (residues 198–250) (PDB code: 7PRE). Structures have been optimally superimposed considering only the N-terminal α-helix (residues 208–221) (right panel) or the C-terminal α-helix (residues 239–246) (left panel). The relative orientation of the two α-helices is loose and only minimally constrained by the interactions between side chains of residues $Val_{215}$, $Ile_{241}$, and $Phe_{229}$ (labelled and colored in green, hydrophobics, and pink, aromatic).

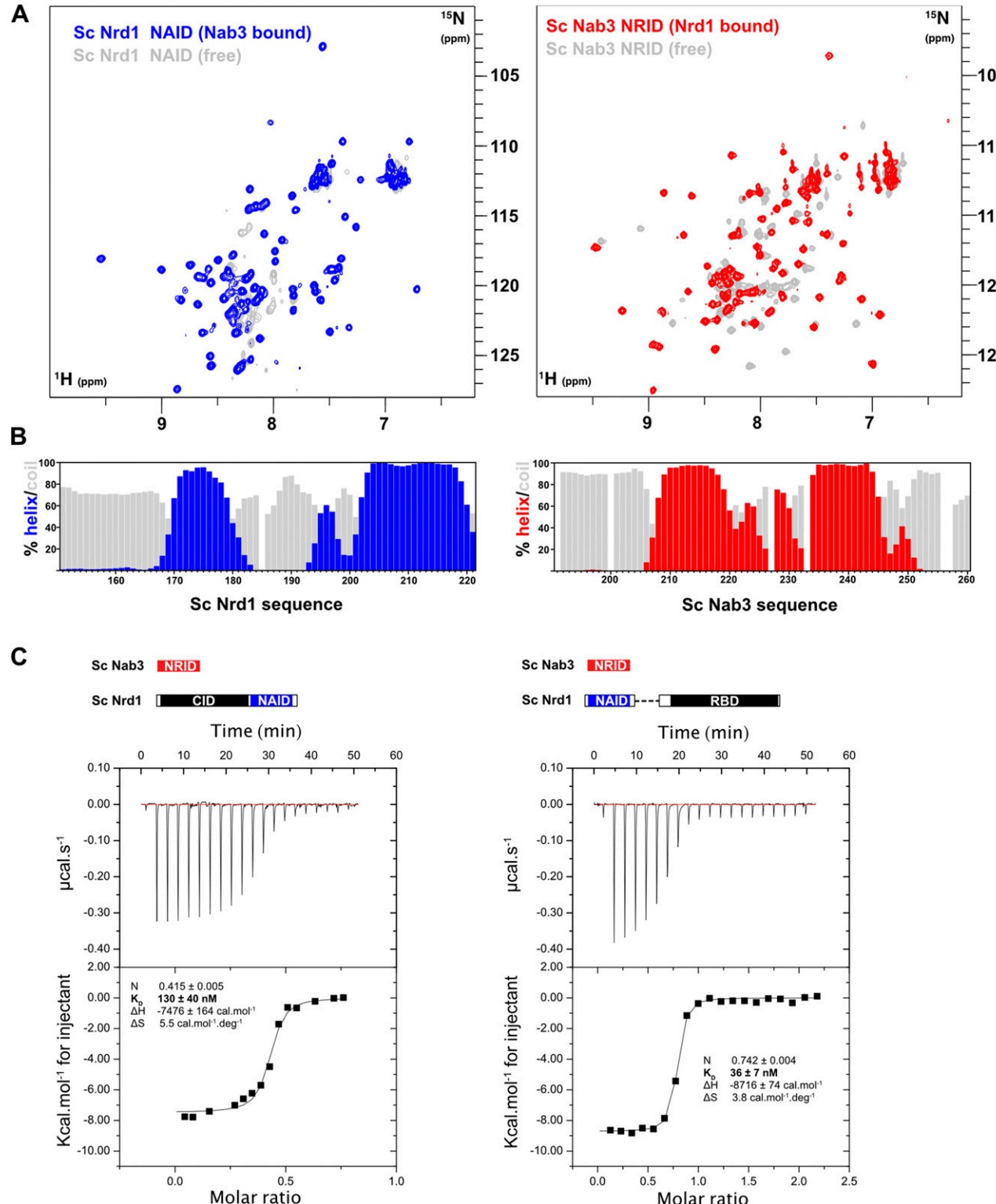

**Figure 2. Nuclear magnetic resonance (NMR) and thermodynamic analysis of Nrd1–Nab3 heterodimerization.**
**(A)** Superposition of the $^1$H-$^{15}$N-HSQC spectra of Nrd1 NAID (residues 147–222, left panel) in its free form (grey) and after addition of unlabelled Nab3 NRID (residues 191–261) (blue). Analogous NMR spectra comparison for $^{15}$N-labelled Nab3 NRID (right panel) showing the superposition of free (grey) and Nrd1 NAID-bound (red) NMR spectra. Unlabelled proteins were added in excess to ensure the saturation of the labelled ones. **(B)** Bar charts showing the per-residue population percentage of α-helix (blue/red) and random coil (grey bars) secondary structure for bound forms of Nrd1 NAID (left panel) and Nab3 NRID (right panel). **(C)** Isothermal titration calorimetry analysis of two different Nrd1–Nab3 interactions. The Nab3$_{191-261}$ construct was titrated over two Nrd1 constructs (left: Nrd1$_{1–222}$ and right: Nrd1$_{147-222/290-489}$) including the domains shown in the scheme. Thermograms (upper panels) and binding isotherms (lower panels) are shown for each titration, together with the equilibrium dissociation

Nrd1 heterodimerization (Fig 3B). The Nab3 segment forms the core of the structure with the Nrd1 acting as a clamping device that fastens Nab3 in a unique conformation. The Nrd1 regions whose NMR signals are missing in the free form organize into two long helices, $Lys_{171}$-$Asp_{180}$ (helix $\alpha1$ in Fig 3B) and $Asn_{201}$-$Lys_{221}$ (helix $\alpha2$), that intimately interact with Nab3 residues. These two helices are separated by a long interconnecting loop that interacts with helix $\alpha2$ and with Nab3 (Fig 3B, left panel). The Nab3 region shows three $\alpha$-helices: $Tyr_{208}$-$Ser_{220}$ (helix $\alpha3$) and $Gln_{234}$-$Ser_{247}$ (helix $\alpha5$) that roughly coincide with those observed in the free form (Fig 1C, right panel and Fig 1E), and a short helix turn $Gln_{228}$-$His_{231}$ (helix $\alpha4$) that was also present in some of the conformers of the free Nab3 NRID structure (Fig 1E).

The long-range $Phe_{229}$-$Ile_{241}$ contact, seen in free Nab3 NRID (Fig 1E), is maintained in the chimera (Fig 3C, left panel), perhaps because it is important to restrict the conformational sampling of Nab3. Nearly all the hydrophobic residues (Phe, Ile, Val, and Leu) are involved in the Nrd1/Nab3 interface of the chimera (Fig 3C), defining a well-ordered core. Many of these residues are totally conserved or at least their hydrophobic character is conserved (Figs 1A and S1A and B). Besides, four of the five methionine and one of the two tyrosine residues (all of these in the Nab3 part) are interfacial. Indeed, the phenolic OH of Nab3 $Tyr_{217}$ is solvent-protected (Fig S6A) and, although we could not identify hydrogen bonds involving this group within the structural ensemble, the spatial proximity of the conserved Nrd1:$Arg_{173}$ and the NOEs between both side chains suggest a possible hydrogen bond interaction (Fig S6B). In addition, the hydroxyl group of Nab3 $Ser_{247}$, that is also detected (thus, protected from solvent exchange) and close to Nab3 $Tyr_{217}$ and Nrd1 $Arg_{173}$ (Fig S6B), might be also involved in that hydrogen bond network.

The relative orientation of the helices in the chimera is further defined by two hydrogen bond networks involving side chains of polar residues (Gln and Asn): Nab3 $Asn_{225}$ and Nrd1 $Gln_{205}$ in one end (Fig 3C left), and Nab3 $Gln_{214}$ and Nrd1 $Gln_{217}$ in the opposite site of the structure, being this later interaction solvent-protected (Fig 3C right). Among these residues, Nrd1 $Gln_{205}$ and Nab3 $Gln_{214}$ are totally conserved (Fig 1A), whereas their partners are more variable but always having polar side chains.

In conclusion, the structure of the Nrd1–Nab3 chimera reveals the atomic details of Nrd1/Nab3 heterodimerization, where hydrophobic interactions and two strategically placed hydrogen bond networks are the critical elements for protein–protein recognition and include most of the evolutionarily conserved residues of both proteins.

### The integrity of the Nrd1/Nab3 interface is crucial for cell survival

Deletion of Nrd1 NAID is not lethal but was shown to cause a strong temperature-sensitive phenotype (Vasiljeva et al, 2008). Now, the reported structure of the Nrd1–Nab3 chimera allows studying the relevance of Nab3/Nrd1 heterodimerization in vivo, by designing mutations that potentially destabilize this interaction, similarly as we did for the Nrd1 RBD (Franco-Echevarría et al, 2017). We used a *LEU* plasmid containing full-length *NRD1* to generate several

mutations in Nrd1 NAID (Fig 4A). Wild-type Nrd1 (wt.) and mutants' plasmids were used to transform a *S. cerevisiae* strain lacking the genomic copy of *NRD1* and expressing it from a centromeric *URA* plasmid. After plasmid shuffling, the resulting wt. and mutant strains were tested for temperature-sensitive phenotypes.

Mutations targeted hydrophobic residues of Nrd1 belonging to helix $\alpha2$ ($Leu_{209}$, $Ile_{213}$, and $Leu_{216}$) and to the extended segment that contacts it ($Leu_{189}$, $Leu_{193}$, and $Leu_{197}$) in the structure (Fig 4B), and were designed to induce mild (Ile/Leu to Ala; Fig 4C) or highly destabilizing effects (Ile/Leu to Arg, Fig 4D) on the Nab3/Nrd1 heterodimer stability. The first set of mutants showed no evident temperature-sensitive phenotype (Fig 4C). The *ndr1*-K335E mutant, located in the RBD and exhibiting slow-growing phenotype at 37°C (Franco-Echevarría et al, 2017), was included as a reference. This set of mutants replaces bulky residues (Leu/Ile) at the hydrophobic core of the Nrd1/Nab3 chimera with a smaller one (Ala), creating energetically unfavourable voids. However, it seems that cells can tolerate these mutations (Fig 4C). Thus, we took a more disturbing approach by mutating to arginine (charged and bulky amino acid) three buried positions of the Nrd1 helix $\alpha2$ ($Leu_{209}$, $Ile_{213}$, and $Leu_{216}$). Surprisingly, neither *nrd1*-L216A and in particular not *nrd1*-L216R mutants showed growth defects compared with *wt* cells (Fig 4D). Perhaps, $Leu_{216}$ is more tolerant to changes due to its terminal location within the Nrd1 NAID. This idea is reinforced when observing the phenotypes of the other two mutants, *nrd1*-I213R and *nrd1*-L209R, in the preceding helix turns of $\alpha2$. The first one clearly shows slow growth at 34$\underline{o}$C and almost thermosensitivity at 37°C; the second one already displays slow growth at 28°C and thermosensitivity at 34°C and 37°C (Fig 4D). The effect of $Leu_{209}$ to $Arg_{209}$ substitution is stronger; indeed, the *nrd1*-L209R growth phenotypes are similar to those shown by the cells where the Nrd1 NAID is completely eliminated (Vasiljeva et al, 2008). Altogether our in vivo results show that even partial perturbation of the Nab3/Nrd1 structure causes an important impact on cell viability, and unveil the functional relevance of the $Leu_{209}$ and $Ile_{213}$ residues. Moreover, our results suggest that the destabilizing effect of these mutations is directional (from inside to outside) along Nrd1 helix $\alpha2$ (*nrd1*-L209R> *nrd1*-I213R> *nrd1*-L216~wt). The in vivo effects of some of these single amino acid substitutions emphasise on the crucial biological role of Nab3/Nrd1 heterodimerization and further demonstrate that the Nrd1–Nab3 chimera is a realistic model of the physiological heterodimer.

# Discussion

### Structural similarities between poly(A)-dependent and NNS transcription termination pathways

*S. cerevisiae* uses two different termination pathways that can operate at various stages during transcription. The activity of the different termination complexes depends on the Rpb1 CTD phosphorylation code and is achieved by proteins containing CIDs of

constant $K_D$(1/$K_B$), enthalpic ($\Delta H$), and entropic contributions ($\Delta S$), and stoichiometry (N) values calculated from data fitting to one-site binding model. Experiments were performed at 15°C.

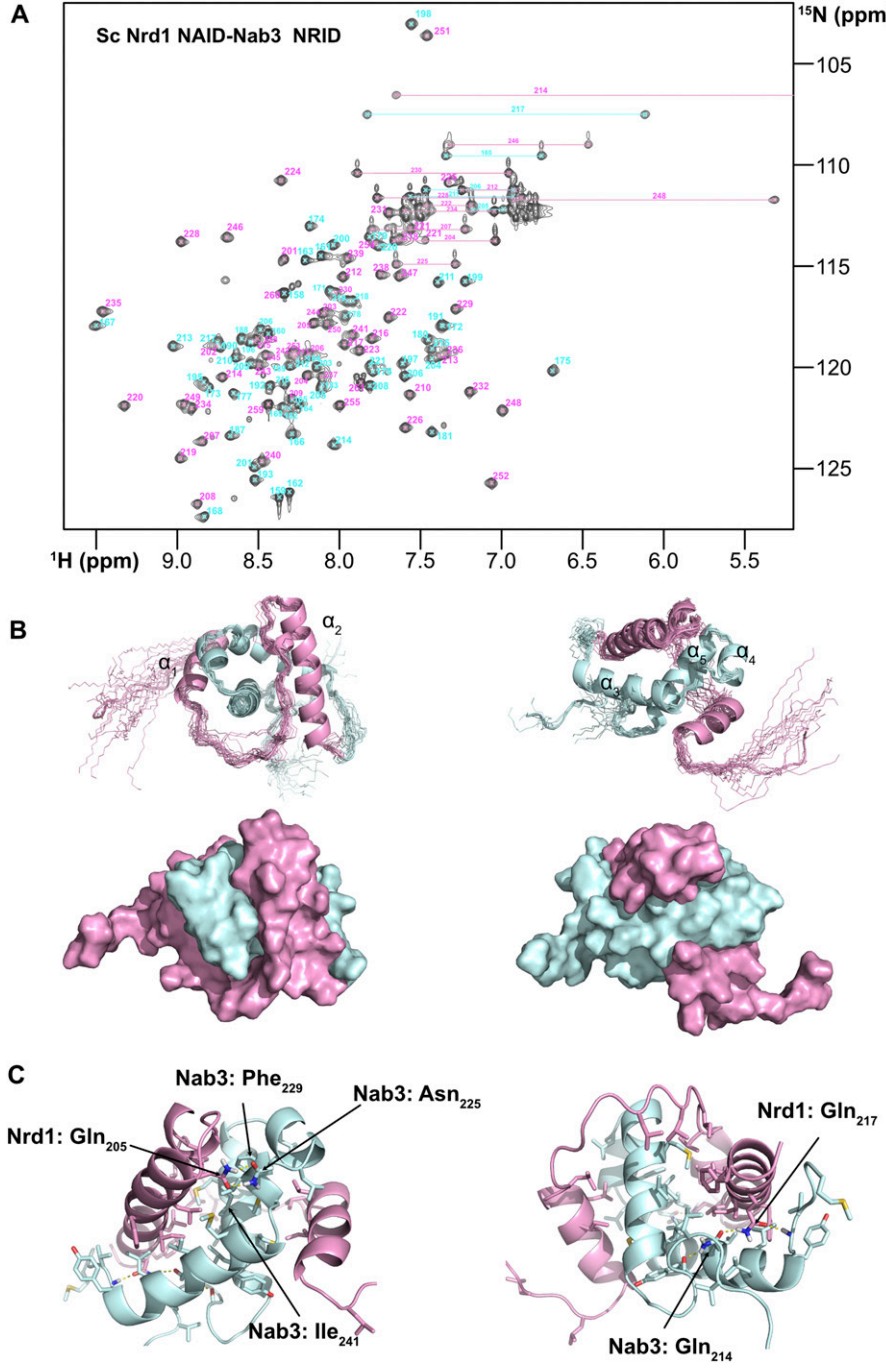

**Figure 3.   Nuclear magnetic resonance structure of the Nrd1–Nab3 chimera.**
**(A)** $^1$H-$^{15}$N HSQC spectrum of the Nrd1$_{158-222}$-Nab3$_{202-261}$ chimera recorded at 800 MHz and 25°C. Cross-peaks assignments have been labelled according to the amino acid sequences of Nrd1 (in pink) and Nab3 (in cyan) fragments that compose the chimera. The horizontal lines mark the two cross-peaks of amide NH$_2$ moieties in side chains of Gln and Asn residues. **(B)** Superposition of the 20 structural models calculated by nuclear magnetic resonance (statistics in Table S1) (upper panels) (PDB code: 7PRD). Two different orientations are shown. The Nrd1 in the chimera is colored in light pink and the Nab3 part in light cyan. Regular secondary structure elements are named consecutively (α-helices α1 to α5). Surface representations of the structure in the two selected orientations and with the same color code are shown below. **(C)** Structural details of the interaction between Nrd1 and Nab3 parts of the chimera. Only side chains of residues involved in heterodimeric contacts are shown. The interface is mainly formed by hydrophobic residues with the exception of Nrd1 Gln$_{205}$ and Gln$_{217}$ with Nab3 Gln$_{214}$ and Asn$_{225}$ that participate in two hydrogen bond networks (yellow dashed lines) that are buried inside the structure. The Nab3 Phe$_{229}$-Ile$_{241}$ contact, present in the free form (Fig 1E), is maintained in the Nrd1–Nab3 chimera. **(B)** Residues have been numbered according to the Nrd1 and Nab3 sequences and colored as in panel (B).

different specificities (Porrua & Libri, 2015). The phosphorylation status of the Rpb1 CTD changes dynamically during transcription (Heidemann et al, 2013). Ser$_5$-P is dominant after transcription starts but becomes progressively less important as it progresses to elongation and termination phases. In contrast, Ser$_2$-P levels show the opposite pattern and become dominant towards the end of the transcription units. Tyr$_1$-P shows a similar profile than Ser$_2$-P, but is erased close to the polyadenylation sites. Pcf11 and Nrd1 have CIDs that specifically recognise the Ser$_2$-P (Meinhart & Cramer, 2004; Lunde et al, 2010)

and Ser$_5$-P (Vasiljeva et al, 2008; Kubicek et al, 2012) peptides, respectively (Fig 5A). In addition, Nrd1 CID can be displaced from the Ser-5 CTD by competitive binding of short segments of Trf4, a component of the TRAMP complex involved in snoRNA precursors (Tudek et al, 2014), and Sen1 (Zhang et al, 2019; Han et al, 2020) that probably helps to disengage the NNS machine from the running transcription complex. Another resemblance between both machineries is the recognition of specific terminator sequences in the transcript, which is attained by two pairs of proteins Nrd1/Nab3 (NNS) and Hrp1/Rna15 (CFI) (Fig 5A). These proteins

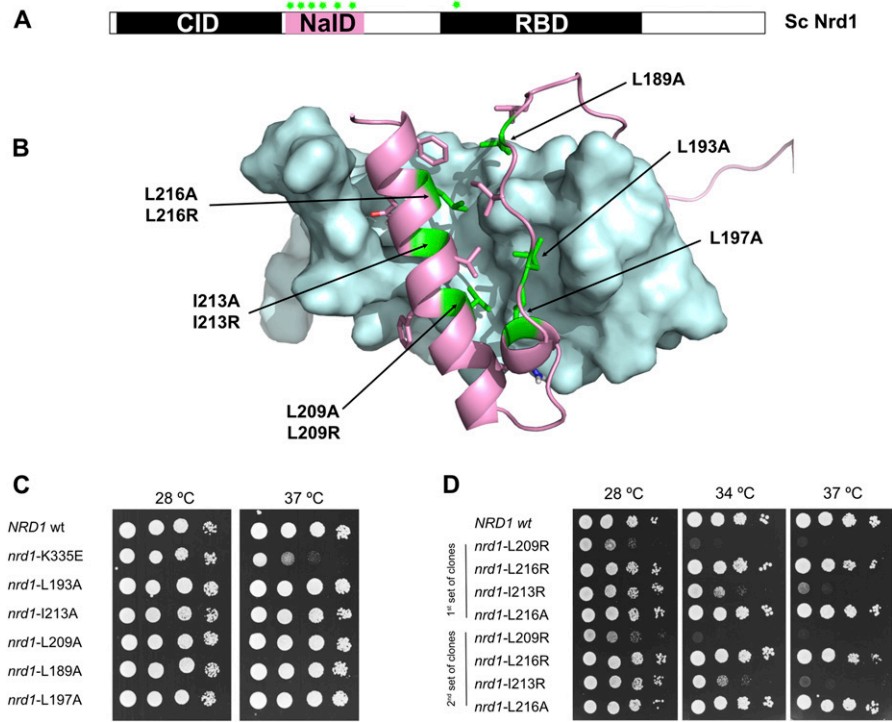

**Figure 4. Functional analysis of Nrd1/Nab3 heterodimerization.**
**(A)** Scheme representing the distribution of the analyzed mutants (indicated as green starts). Six positions in Nrd1 NAID domain were mutagenized (see specific details in the text). **(B)** The six mutagenized residues in Nrd1 NAID correspond to hydrophobic amino acids ($Leu_{189}$, $Leu_{193}$, $Leu_{197}$, $Leu_{209}$, $Ile_{213}$, and $Leu_{216}$) buried in the structure. These Leu or Ile side chains were replaced with Ala (conservative mutation) or Arg (disrupting mutation). **(C, D)**. Analysis of the growth phenotypes of the *nrd1* mutants and wild-type cells (wt.). The temperature-sensitive mutant *nrd1*-K335E, previously identified in the RNA-binding domain (Franco-Echevarría et al, 2017), is included as reference. Cultures were serially diluted (1/10), spotted on selective SC media plates and grown at the indicated temperatures for 2–3 d. **(C)** The first set of mutants (Leu/Ile to Ala) does not show differential behavior compared to wt. at the two tested temperatures. In comparison, the *nrd1*-K335E temperature-sensitive mutant shows the expected growth phenotype at 37°C (Franco-Echevarría et al, 2017). **(D)** Among the second set of mutants, including Leu/Ile to Arg mutations, *nrd1*-L209R and *nrd1*-I213R show strong growth defects, even lethality at 34°C and 37°C for *nrd1*-L209R mutant. Two clones of each mutant were tested.

contain RRMs that achieve RNA sequence specificity by working together to recognise segments of single-strand RNA near the termination sites. To accomplish this cooperative recognition, RBPs have to bind to the RNA as a single entity. Nrd1 and Nab3 form a heterodimer, whose structural features have been described in this work, whereas Rna14 acts as scaffold for Rna15 and Hrp1 (Fig 5A). The Hrp1/Rna14 interaction has been mapped to Hrp1 RRMs by NMR (Barnwal et al, 2012), but the structural details remain unknown. On the other hand, the Rna15/Rna14 heterodimer involves the so-called hinge and Monkeytail domains (Moreno-Morcillo et al, 2011) with Rna14 wrapping around a bundle of helices of Rna15 (Fig 5A and B). This binding mode is strikingly similar to the Nrd1/Nab3 one described in our chimera (Fig 5B), where Nrd1 wraps around the bundle of helices of Nab3. Although both complexes do not superimpose and many structural differences can be found, their protein–protein recognition strategy is similar. In the Rna15/Rna14 heterodimer, both the hinge (from Rna15) and the Monkeytail (from Rna14) domains appear to be unfolded in their free states (Moreno-Morcillo et al, 2011). In contrast, in the Nrd1/Nab3 there is some level of pre-structural arrangement, at least in Nab3, which probably alleviates the entropic cost of the heterodimer formation. Furthermore, the surface buried by the Rna14/Rna5 complex (4,900 ± 200 $\mathring{A}^2$ [Moreno-Morcillo et al, 2011]) is larger than that calculated for the Nab3/Nrd1 heterodimer (3,364 ± 95 $\mathring{A}^2$). In this context, a recent statistical study shows that buried interfaces contribute between 3 and 4 cal $mol^{-1}$ $\mathring{A}^{-2}$ to the free energy (Chen et al, 2013). In the case of the Nab3/Nrd1, this would lead to theoretical ΔG of −10.1 to 13.5 kcal $mol^{-1}$, which is slightly lower than the −9.8 kcal $mol^{-1}$ value obtained by ITC (Fig 2C, right panel), showing that the amount of buried surface is in reasonable agreement with the heterodimerization energetics.

## Is Nrd1/Nab3 heterodimerization conserved within the fungal kingdom?

The structural comparison between the two transcription termination complexes in *S. cerevisiae* shows interesting parallelisms. Nrd1 presents a unique architecture within the NNS machinery, comprising a CID, a heterodimerization domain, and an RBD. The structure of the RBD (Franco-Echevarría et al, 2017) and the reported Nab3-Nrd1 chimera structure (a faithful model of the heterodimer) are exclusive of Nrd1-like proteins. The search for Nrd1 orthologs (https://omabrowser.org/) found 121 fungal sequences; there are not Nrd1-like proteins in other kingdoms of life. Besides, these Nrd1-like proteins showed clear conservation patterns when looking at the RBD and CID domains (data not shown). In contrast, Nrd1 NAID is well conserved within the *Saccharomyces* clade (Figs S1A and B and S7) but no in other fungal species which show large insertions between the two helices. These differences would likely affect the Nrd1/Nab3 heterodimer architecture and perhaps even compromise its formation. Even the evolutionary-close *Candida* clade showed significant differences in this region (Fig S7), suggesting that the Nrd1/Nab3 heterodimer might be an exclusive feature of the *Saccharomyces* clade. In support of this hypothesis, experimental data show that *Schizosaccharomyces pombe* Seb1, Yas9, and Dbl8, orthologs of *Saccharomyces cerevisiae* Nrd1, Nab3, and Sen1, respectively, do not form a stable complex (Lemay et al, 2016). Even more, these proteins are not involved in transcriptional termination of snRNA genes, suggesting that the NNS-dependent termination does not exist in fission yeast (Larochelle et al, 2018). With this evidence, and in conjunction with the evolutionary data (Fig S7), it is tempting to speculate that the emergence of heterodimerization

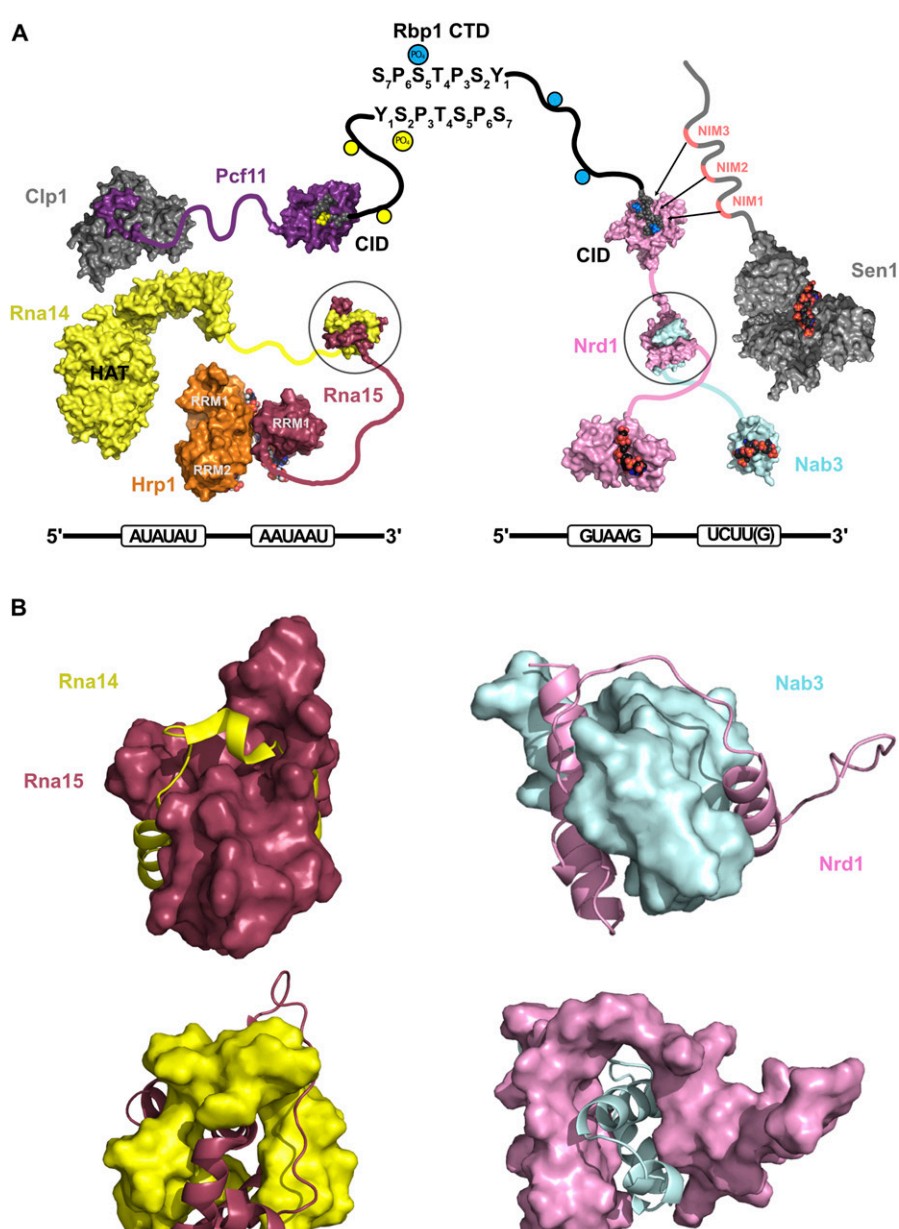

**Figure 5. Structural comparison between CFI and NNS complexes.**
**(A)** The structural models depict the current knowledge about the organization and interactions within the Cleavage Factor I and Nab3–Nrd1–Sen1 complexes, that are involved in the two transcription termination pathways in yeast (see the Introduction section for details). On the right, termination of short transcripts is associated to Ser$_5$ phosphorylation mark in RNA Pol II (blue dots in the schematic representation of Rpb1 CTD) that are recognized by Nrd1 CID (PDB: 2IO6 in pink and Rpb1 CTD in grey/blue [Ser$_5$-P]). On the nascent transcript, Nrd1 (PDB: 5O1Y in pink) and Nab3 (PDB: 2L41 in cyan) RNA-binding domains recognize specific terminator sequences (black line and boxed RNA sequences below). The helicase Sen1 (PDB: 5MZN) also recognizes unspecific RNA sequences, and its intrinsically disordered region contains three Nrd1 interaction motifs (NIMs): NIM1, NIM2, and NIM3 (marked in red) that can interact with the CID, competing out the Rpb1 CTD and allowing the termination process to evolve to its final steps (Zhang et al, 2019; Han et al, 2020). On the left, CFI uses similar strategies. The CID of Pcf11 (PDB: 1SZA in purple) recognizes Ser$_2$-P CTD-derived peptides (yellow dots and yellow atoms in the 1SZA structure), typical of long-elongated transcripts, whereas Hrp1 (orange) and Rna15 (maroon) (PDB: 2KM8) recognize the polyadenylation signal and enhancement elements. Clp1 (grey) recognizes a Pcf11 peptide (in purple) (PDB: 2NPI) and also interacts with other proteins of CFI (yet-unknown structures). The Rna14 HAT domains (yellow) interact with Hrp1 RRMs (Barnwal et al, 2012) and its Monkeytail domain forms a heterodimer with the C terminus or Rna15 (maroon) (PDB: 2L9B). This heterodimer has a similar structure as the Nrd1–Nab3 chimera (PDB: 7PRD this work). **(B)** Comparison between the structures of Rna14/Rna15 heterodimer and Nrd1–Nab3 chimera. In both cases, models have been represented as a surface/ribbon mixture for each of the components, and alternating between them in top and bottom figures (identical orientation for each structure). Rna14 Monkeytail domain (yellow) and Nab3 interacting domain in Nrd1 (pink) wrap around their partners in a similar way, creating large protein–protein interfaces. In the structures, Rna15 (maroon) and Nab3 (cyan) form compact helix bundles.

between the two RBPs (Nrd1-like and Nab3-like) was the critical molecular event that triggered the development of a new transcription termination mechanism, specialized in small non-coding RNAs, in the *Saccharomyces* clade.

# Materials and Methods

### Circular dichroism measurements

CD spectra were recorded on a Jasco J-810 spectropolarimeter in pure water at 25°C and using a 0.1-cm path-length cell for far-UV measurements. Experiments were acquired with a scan speed of 50 nm min$^{-1}$, a response time of 4 s and a 0.5-nm band width. Protein concentrations were 16 $\mu$M for Nrd1$_{147-222}$ and 20 $\mu$M for Nab3$_{191-261}$.

### Protein expression and purification

Nrd1 and Nab3 sequences were amplified from *Saccharomyces cerevisiae* genomic DNA (Novagen) using specific DNA primers (Macrogen) and high fidelity KOD DNA polymerase (Novagen). The fragments were cloned into a pET28-modified vector encoding TxA-6xHis-TEV cleavage site as a N-terminal fusion cassette (TxA correspond to the *E. coli* thioredoxin A sequence). Nrd1, Nab3, and

chimeric Nrd1–Nab3 constructs were obtained and overexpressed in *E. coli* BL21(DE3) cells. Cells were grown in Luria-Bertani (LB) broth for natural abundance samples, and in KMOPS minimal media (Neidhardt et al, 1974) for $^{15}N/^{13}C$ labelled samples. In the latter case, labelled ammonium chloride or glucose as (Cambridge Isotope Laboratories) sole nitrogen and carbon sources were used. Natural abundance and isotopically labelled cultures were induced at $OD_{600}$ = 0.6–0.8 with 0.5 mM IPTG (Sigma-Aldrich) at 25°C (or 16°C) for 12 h (or 20 h) and then harvested and frozen at –20°C until further use. For selective $^{13}C$-methyl labelling, cultures were grown in $^{15}N$-KMOPS minimal media until $OD_{600}$ = 0.3–0.4 and then supplemented with $\alpha$-ketobutyric acid ($^{13}C$-methyl) (120 mg/l) and $\alpha$-ketoisovaleric acid ($^{13}C$-methyl) (70 mg/l) (Cambridge Isotope Laboratories) adapting previously reported protocols (Goto et al, 1999).

Resuspended cell pellets (in buffer A: 25 mM potassium phosphate pH 8.0, 300 mM NaCl, 10 mM imidazole, 5 mM $\beta$-mercaptoethanol, and 1 tablet/50 ml of EDTA-free protease inhibitors [Roche]) were sonicated, centrifuged and the supernatant filtered through a 0.22-$\mu$m filter prior loading into a HisTrap 5 ml column (GE Healthcare). The IMAC (immobilized metal affinity chromatography) column was washed with buffer B (25 mM potassium phosphate, pH 8.0, 500 mM NaCl, 30 mM imidazole, and 5 mM $\beta$-mercaptoethanol) and eluted with buffer C (25 mM potassium phosphate, pH 8.0, 300 mM NaCl, 300 mM imidazole, and 5 mM $\beta$-mercaptoethanol). The samples were exchanged to buffer A by desalting chromatography (G-25 resin) or dialysis and 100 $\mu$g/ml of homemade TEV protease were added prior overnight digestion at 16°C. Undigested fusion protein, cleaved tag, TEV, and some other impurities were removed by a second IMAC chromatography, using the same buffers as before, and the target protein was collected in the flow-through (buffer A) or buffer B fractions (depending on the protein construct). Next, the protein samples were concentrated by ultrafiltration (Vivaspin 10 kD cut off membrane), followed by gel filtration with a Superdex 200 10/300 GL column (GE Healthcare). Finally, samples were exchanged to their final buffer, depending on the subsequent experiments, and their purity checked by PAGE–SDS.

## NMR

The concentration of the different protein constructs was determined from the aromatic contribution to the UV spectrum at 280 nm, with the exception of Nrd1$_{147-222}$ that lacks this type of residues and absorbance measurements at 205 nm were used to estimate the concentration (Anthis & Clore, 2013). NMR samples were prepared at concentrations ranging 100–1,000 $\mu$M in buffer containing 25 mM potassium phosphate, pH 6.6, 25 mM NaCl, 1 mM DTT, and 10% $D_2O$. NMR assignments of Nrd1$_{147-222}$, Nab3$_{191-261}$ in their free and bound forms were obtained from triple-resonance backbone experiments 3D HNCA, HNCO, CBCA(CO)NH, and HNCACB (Sattler et al, 1999) recorded at 25°C on Bruker AV800 and AV600 spectrometers, both with triple-resonance cryoprobes. For the structure calculation of Nab3$_{191-261}$, two 2D NOESY spectra (in 10% and 100% $D_2O$) were acquired in a Bruker AV800 spectrometer with 480 $\mu$M samples and 80 ms mixing time.

For the Nrd1–Nab3 chimera, we first obtained the assignments of the Nrd1$_{147-222}$-Nab3$_{202-261}$ construct using 3D HNCA, HNCO,

CBCA(CO)NH, and HNCACB triple-resonance backbone experiments, and also 3D HcCH-TOCSY, hCCH-TOCSY experiments (Sattler et al, 1999) recorded on a Bruker AV600. The $^1H$, $^{15}N$, and $^{13}C$ assignments of the optimized chimera, Nrd1$_{158-222}$-Nab3$_{202-261}$, were easily transferred from the previous set of data and confirmed with 3D HNCA, HNCO, CBCA(CO)NH, HcCH-TOCSY, and hCCH-TOCSY spectra. NMR experiments of that optimal chimeric construct were recorded in 10 mM sodium acetate (D3, 99%), pH 5.1, 25 mM NaCl, and 1 mM DTT buffer. NOE-derived distance restraints were obtained from five different NOESY-type experiments: 2D NOESY ($H_2O/D_2O$ 9:1), 2D NOESY ($D_2O$), 3D $^1H$-$^{15}N$-HSQC-NOESY, $^1H$-$^{13}C$-HSQC-NOESY, and $^1H$-$^{13}C$-HSQC-NOESY-$^1H$-$^{13}C$-HSQC (Sattler et al, 1999). The last two spectra were recorded on $^{13}C$-methyl selectively labelled Leu, Val and Ile ($\delta1$) samples. All these spectra were recorded at 25°C on a Bruker AV800 spectrometer, with ~1 mM protein concentration and 60 ms mixing time. Backbone angle restraints were obtained from $^{13}C$ and $^1H$ chemical shifts with TALOS+ (Shen et al, 2009). Structures were calculated with CYANA 3.0 (Güntert & Buchner, 2015) by a standard simulated annealing protocol starting from 50 random conformers (statistics in Table S1). The 20 lowest target function conformers were selected as representative of the NMR structure. NMR data were handled and analyzed with Topspin (Bruker), and ccpnNMR Analysis (v2) software (Vranken et al, 2005), and the structures were visualized with Pymol (DeLano Scientific LLC).

## ITC

Experiments were carried out on a MicroCAl iTC200 (Malvern Instruments) at 15°C in 20 mM potassium phosphate (pH 7.0), 150 mM NaCl, and 1 mM $\beta$-mercaptoethanol. In all cases concentrated Nab3$_{191-261}$ (198 $\mu$M) in the syringe, was titrated into Nrd1 variants: Nrd1$_{147-222/290-489}$ (19 $\mu$M), Nrd1$_{1-222}$ (28 $\mu$M), and txAHTEV-Nrd1$_{147-222}$ (54 $\mu$M). Experiments were performed in duplicate with injections of 2 $\mu$l (0.4 $\mu$l for first point) separated by 150 s delays to recover thermal power baseline and continuous stirring in the cell (1,000 rpm) for correct mixing. The reference cell was filled with water in all the experiments. Data were processed by removing the blank experiment (dilution of Nab3$_{191-261}$ in buffer) and adjusted to one-site binding model with Origin 7.0 (OriginLab).

## *S. cerevisiae* strains and mutants

*NRD1* mutations were introduced in a centromeric *LEU* pRS415-*NRD1* plasmid by QuickChange mutagenesis (Agilent) using specific DNA oligonucleotides (Macrogen). The corresponding yeast strains were constructed following the procedures reported in our previous work (Franco-Echevarría et al, 2017). Wild-type and mutant plasmids were used to transform EJS101-9d strain (*Mat a, ura3-52, leu2-3,112, trp1-1, his3-11,15, ade2-1, met2Δ1, lys2Δ2, can1-100,* and *nrd1::HIS3* [pRS316-*NRD1*] [Steinmetz & Brow, 1996]) that lacks the genomic *NRD1* gene and expresses it from a centromeric *URA* pRS316-*NRD1* plasmid (*NRD1* is required for *S. cerevisiae* viability). Transformants were selected in URA-LEU medium and then grown in 5-FOA containing medium to enable the selective loss of pRS316-*NRD1* and expression of *NRD1* (wt and mutant genes) from the *LEU* plasmids. None of the obtained mutant strains were lethal, and therefore we grew them at different temperatures to evaluate

potential growth defects. For that purpose, we performed serial dilution assays (1:10) of the corresponding yeast strains on selective medium plates and grown them for 2–3 d at the indicated temperatures. Prof S Buratowski kindly provided the original yeast strain (EJS101-9d) and the pRS415-*NRD1* plasmid.

## Data Availability

Atomic coordinates have been deposited in the Protein Data Bank (PDB) under the accession codes 7PRE (Nab3$_{191-261}$) and 7PRD (Nrd1$_{158-222}$-Nab3$_{202-261}$), and 1H/15N and 13C chemical shifts under the Biological Magnetic Resonance Data Bank (BMRB) accession numbers 34669 (Nab3$_{191-261}$) and 34668 (Nrd1$_{158-222}$-Nab3$_{202-261}$).

## Supplementary Information

## Acknowledgements

NMR experiments were performed in the "Manuel Rico" NMR laboratory (LMR) of the Spanish National Research Council (CSIC), a node of the Spanish Large-Scale National Facility (ICTS R-LRB). Funding was provided by grants: PID2020-112821GB-I00 to JM Pérez-Cañadillas and MÁ Jiménez funded by MCIN/ AEI /10.13039/501100011033/; CTQ2017-84371-P to JM Pérez-Cañadillas and MÁ Jiménez funded by MCIN/ AEI /10.13039/501100011033/ and by "ERDF A way of making Europe"; BFU2017-84694-P to O Calvo funded by MCIN/ AEI /10.13039/501100011033/ and by "ERDF A way of making Europe"; and RED2018-102467-T to O Calvo and JM Pérez-Cañadillas funded by MCIN/ AEI /10.13039/501100011033/. JM Pérez-Cañadillas was also funded by a grant of the Biomedicine program of Community of Madrid (B2017/BMD-3770 RYPSE-CM) that is co-financed with ERDF and ESFESF. The IBFG is supported in part by an institutional grant from the "Junta de Castilla y León" (Programa "Escalera de Excelencia" de la Junta de Castilla y León, Ref. CLU-2017-03 co-funded by O.P. ERDF from Castilla y León 14-20). JM Pérez-Cañadillas would like to thank to Felipe Pozo Lucas for the design and construction of the RYPSE-CM project web page.

### Author Contributions

B Chaves-Arquero: data curation, investigation, methodology, constructed plasmids, expressed and purified proteins, performed NMR experiments, analyzed NMR data, and calculated the 3D structures.
S Martínez-Lumbreras: data curation, validation, investigation, methodology, analyzed NMR data, and calculated the 3D structures.
S Camero: investigation, methodology, constructed plasmids, expressed and purified proteins, and performed and analyzed CD experiments.
CM Santiveri: investigation and obtained and analyzed ITC experiments.
Y Mirassou: constructed plasmids and expressed and purified proteins.
R Campos-Olivas: investigation and obtained and analyzed ITC experiments.
MÁ Jiménez: funding aquisition, investigation and performed NMR experiments.
O Calvo: investigation and experiments with *S. cerevisiae* strains and mutants

JM Pérez-Cañadillas: conceptualization, data curation, formal analysis, supervision, funding acquisition, validation, investigation, visualization, writing—original draft, review, and editing, constructed plasmids, expressed and purified proteins, performed NMR experiments, analyzed NMR data, calculated the 3D structures, and conceived project. .

### Conflict of Interest Statement

The authors declare that they have no conflict of interest.

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
