## [Reviewer comments · Life Science Alliance]

Life Science Alliance

Structural basis of Nrd1-Nab3 heterodimerization

Belén Chaves-Arquero, Santiago Martínez-Lumbreras, Sergio Camero, Clara Santiveri, Yasmina Mirassou, Ramón Campos-Olivas, María Jiménez, Olga Calvo, and José Perez-Cañadillas

DOI: <https://doi.org/10.26508/lsa.202101252>

Corresponding author(s): José Perez-Cañadillas, Institute of Physical-Chemistry

Review Timeline:

Submission Date:	2021-10-04
Editorial Decision:	2021-10-29
Revision Received:	2021-11-23
Editorial Decision:	2021-12-09
Revision Received:	2021-12-14
Accepted:	2021-12-15

Scientific Editor: Novella Guidi

Transaction Report:

October 29, 2021

Re: Life Science Alliance manuscript #LSA-2021-01252-T

Dr. Jose M Pérez-Cañadillas
CSIC

Dear Dr. Pérez-Cañadillas,

Thank you for submitting your manuscript entitled "Structural basis of Nrd1-Nab3 heterodimerization" to Life Science Alliance. The manuscript was assessed by expert reviewers, whose comments are appended to this letter. As you will note from the reviewers' comments below, all reviewers are quite happy with the work and raise few minor concerns. Reviewer 1 is concerned on the lack of description of the chimeric protein, suggesting adding a section describing the generation and validation of the chimeric protein suitable for structural analysis. Reviewer 2 thinks that the data is of high quality and the interpretations are reasonable. The main criticism is of the extent to which much new insight is offered in the report and suggest few minor changes. Reviewer 3 just points out to a wrong citation in the introduction and to remove an overstating sentence in the discussion. We, thus, encourage you to submit a revised version of the manuscript back to LSA that responds to all of the reviewers' points.

Thank you for this interesting contribution to Life Science Alliance. We are looking forward to receiving your revised manuscript.

Sincerely,

B. MANUSCRIPT ORGANIZATION AND FORMATTING:

Reviewer #1 (Comments to the Authors (Required)):

The manuscript by Chaves-Arquero et al describes the solution structure of a chimeric protein mimicking the heterodimer between the yeast proteins Nrd1 and Nab3, involved in transcription termination.

Overall, the manuscript is well presented and scientifically sound.

My only concern is the lack of description of the chimeric protein. Using chimeric proteins to study heterodimer is a powerful tool but the chances of success are generally low without prior knowledge of the structure and therefore the distance between the N- and C-terminal regions of each proteins. In the result section, the authors state "after optimization, we chose the construct Nrd1-158-221-Nab3-203-261". It would be interesting to know how the optimization was done. How many constructs were tested and how were they tested (CD? NMR?). Have the authors also tried to include Nab3 at the N-terminus of the chimera? It seems that there is no linker between the 2 proteins, suggesting that the C-terminal amino acid of Nrd1 is very close in space to the N-terminal amino acid of Nab3. Was this known when generating the chimera or was it just luck?. Did the authors try inserting a linker between the two proteins?

I believe that a section describing the generation and validation of the chimeric protein suitable for structural analysis would benefit the readers.

Minor comments:

Introduction: please replace "tightly regulated by post-transcriptional modifications such as phosphorylation of serines 2, 5, and 7" by "tightly regulated by post-translational modifications such as phosphorylation of serines 2, 5, and 7"

Results section Nrd1-Nab3 heterodimerization:

NMR investigation of the heterodimer was done using the construct Nrd1-148-222, but the ITC experiments were done using Nrd1-1-122 and Nrd1-148-222/290-489. This is a bit strange. Why not doing ITC using the shortest Nrd1-148-222 construct? Could the authors justify the different constructs used in NMR and ITC experiments?

Supplementary Figure 4:

I believe that it would be more informative to show the HSQC overlays in one single figure with the chimera in grey, bound Nrd1 in blue and bound Nab3 in red. That will show better which peaks belong to each protein and which peaks are different. It would also be good to have a plot of the chemical shift difference as a function of the amino acid sequence, to highlight regions similar or different.

Supplementary Table 1:

Could the authors add a line stating how many Noes were used between amino acids of Nrd1 and Nab3 (pseudo intermolecular Noes)?

Results: The integrity of the Nrd1/Nab3 interface is crucial for cell survival

To validate the heterodimeric structure, the authors mutated hydrophobic residues of Nrd1. Why didn't the author mutate Gln205 and 217 that are described as involved in intermolecular hydrogen bonds and are conserved.

Reviewer #2 (Comments to the Authors (Required)):

Perez-Canadillas and colleagues report a structural analysis of the dimerization interaction between yeast RNA binding proteins Nab3-Nrd1. They use a number of biophysical techniques, predominantly NMR, to assess the flexibility and extent of order induced upon dimerization of the isolated Nrd1-binding domain of Nab3 and Nab3-binding domain of Nrd1. They conclude that

the interaction interfaces results from packed helices and induced structure. The authors suggest that the structural arrangement of these two proteins is analogous to RNA14/15 assembly seen for the other major termination mechanism in budding yeast.

This study is a straightforward analysis that continues the probing of the solution structure of Nab3 and Nrd1 by these authors and the Stefl lab. The data is of high quality and the interpretations are reasonable. The main criticism is of the extent to which much new insight is offered in the report. The biological significance of some of the findings: the possible 2:1 stoichiometry between the proteins, the suggestion that the CID "destabilizes" the interaction, the use of the fused Nrd1-Nab3 domains, and the internally deleted Nrd peptide, is questionable. The Discussion is interesting but does not really result from the investigation reported here which is of more limited scope. In sum, the paper is satisfactory but incremental.

Minor comments:

- Number the pages please
- grammatical proofreading is needed
- what is meant by the Nab3 NRID "optimized" construct? Optimized for what?
- Define the acronyms in the Introduction: NMR, CD, ITC
- Typo: in results KD not KDs
- Typo in Supp Fig legend 3: ITC not ICT

Reviewer #3 (Comments to the Authors (Required)):

This is a very nice and simple paper that solves the structure of the Nrd1/Nab3 heterodimerization domains. Starting with a basic analysis of the individual proteins, the authors show that the Nab3 domain forms a small domain consisting of a few helices, and that Nrd1 folds around the Nab3 fold. I'm not an NMR specialist, but the structure looks very plausible and convincing. To solve a technical problem with the proteins, they used a clever trick of fusing the two domains together, but the spectra and other data argue that their proposed structure exists in the context of the full length proteins. In particular, in vivo complementation experiments show that mutations predicted to disrupt the dimer interface lead to growth defects. The discussion points out an interesting structural similarity of the Nrd1/Nab3 dimerization structure to that of the polyadenylation factors RNA14/RNA15, although this clearly reflects convergent evolution rather than a shared ancestor. Overall, I have only two small comments related to references:

1. In the first paragraph of the introduction: I don't think the 2011 Marquardt paper is the right reference for showing that there are two pathways for termination. This paper is really more about degradation pathways. A better reference would be Kim et al. (2006) Mol Cell 24:723. Before this paper, it was less clear that the pA pathway and NNS pathway were really separate, rather than variations of the same mechanism.
2. In the discussion, I think it's overstating things to state that Tyr1 phosphorylation acts to block termination in the middle of genes. That was the model proposed in the Mayer 2012 paper based on circumstantial evidence (ChIP of Tyr1-P and peptide binding). However, mass spec experiments on the CTD find so little tyrosine phosphorylation on the CTD that it can't possibly be enough to block Ser5P and Ser2P binding proteins. I would just take out that sentence, as it's not necessary for the story presented here.

Dear Dr Guidi (Scientific Editor of Life Science Alliance),

Thank you for giving us the opportunity to submit the revised version of our manuscript. We would also like to thank to the three reviewers for their comments and suggestions. Here we send a specific point-by-point answer to their comments.

Sincerely,

Jose M. Perez Cañadillas

Reviewer #1 (Comments to the Authors (Required)):

The manuscript by Chaves-Arquero et al describes the solution structure of a chimeric protein mimicking the heterodimer between the yeast proteins Nrd1 and Nab3, involved in transcription termination.

Overall, the manuscript is well presented and scientifically sound.

My only concern is the lack of description of the chimeric protein. Using chimeric proteins to study heterodimer is a powerful tool but the chances of success are generally low without prior knowledge of the structure and therefore the distance between the N- and C-terminal regions of each proteins. In the result section, the authors state "after optimization, we chose the construct Nrd1-158-221-Nab3-203-261". It would be interesting to know how the optimization was done. How many constructs were tested and how were they tested (CD? NMR?). Have the authors also tried to include Nab3 at the N-terminus of the chimera? It seems that there is no linker between the 2 proteins, suggesting that the C-terminal amino acid of Nrd1 is very close in space to the N-terminal amino acid of Nab3. Was this known when generating the chimera or was it just luck?. Did the authors try inserting a linker between the two proteins?

I believe that a section describing the generation and validation of the chimeric protein suitable for structural analysis would benefit the readers.

We would like to thank the reviewer for her/his comments and suggestions. The design/optimization of the chimeric protein was important in the project, and following the reviewer suggestion we have added a full description of that process in the Results section (Pag 8), accompanied by a new version of Supplementary Figure 4.

We tested four Nrd1-Nab3 constructs and compared their NMR spectra with those of the Nrd1/Nab3 bound forms in the heterodimer. We did not try Nab3-Nrd1 chimeras because the first tested construct, Nrd1₁₄₇₋₂₂₂-Nab3₂₀₂₋₂₆₁, reproduced well the heterodimer sub-spectra (new Supplementary Figure 4A). This construct was based on sequence conservation patterns and we were a bit lucky getting a favorable geometry. However, we have previous experience with other projects where the two configurations of the chimera were necessary. We tested a chimeric construct with a flexible linker, Nrd1₁₆₈₋₂₂₂-GGGSGGGTGGGTGGGS-Nab3₂₀₃₋₂₅₄, but it showed spectral heterogeneity (minor peaks next to the original ones) and a poorer comparison with the heterodimer sub-spectra (new Supplementary Figure 4B). The version with a shorter N-terminus, Nrd1₁₆₈₋₂₂₂-Nab3₂₀₂₋₂₆₁, also shows higher chemical shift differences with the heterodimer (new Supplementary Figure 4B) and spectral heterogeneity. Finally, the last version of the chimera, Nrd1₁₅₈₋₂₂₂-Nab3₂₀₂₋₂₆₁ showed a nearly identical spectrum than the original one, and without spectral heterogeneity.

As the first construct was as good as the last one, and including a linker between the two proteins did not lead to any advantage, we decided not to include the description of the chimera optimization in the manuscript. However, we hope that the reported description benefits the article and is valuable for other researchers following similar approaches to study intermolecular interactions.

Minor comments:

Introduction: please replace "tightly regulated by post-transcriptional modifications such as phosphorylation of serines 2, 5, and 7" by "tightly regulated by post-translational modifications such as phosphorylation of serines 2, 5, and 7"

Change included in the revised version of the manuscript.

Results section Nrd1-Nab3 heterodimerization:

NMR investigation of the heterodimer was done using the construct Nrd1-148-222, but the ITC experiments were done using Nrd1-1-122 and Nrd1-148-222/290-489. This is a bit strange. Why not doing ITC using the shortest Nrd1-148-222 construct? Could the authors justify the different constructs used in NMR and ITC experiments?

We used the txAHTEV-Nrd1₁₄₇₋₂₂₂ instead of Nrd1₁₄₇₋₂₂₂ to get a more accurate estimation of the protein concentration. Nrd1₁₄₇₋₂₂₂ has a near zero extinction coefficient (only one Cys with negligible contribution to absorbance at 280 nm) and Nab3₁₉₁₋₂₆₁ has a very low extinction coefficient at 280 nm (two contributing tyrosines). We thought that potentially large errors in the concentration estimation of both titrant (Nab3) and titrand (Nrd1) could combine to render unreliable data. This is a relatively less important issue in the NMR titration because we aimed to saturate the labeled component (to observe signals corresponding to the pure bound-form) and, therefore, worked in excess of the unlabeled partner.

We should also note that the ITC experiments were performed at slightly different buffer (higher ionic strength) and temperature (15 °C vs 25 °C) than the NMR ones. This was because the first set of ITC experiments that we did, with Nrd1₁₄₇₋₂₂₂ (untagged) and Nab3₁₉₁₋₂₆₁, and at the same temperature (25 °C) than the NMR titrations, showed that the released heat per injection (i.e. binding enthalpy) was close to zero (see figure below). We repeated the experiments several times, until we ran out of Nrd1 sample, and obtained the same result. At that point, we were unsure if the obtained results were due to suboptimal experimental conditions and/or to errors in the determination of the protein concentrations. For this reason, we decided to perform the subsequent ITC experiments at 15°C (as in Vasiljeva *et al.* Nat. Struct. Mol. Biol. 2008) and with the txAHTEV-Nrd1₁₄₇₋₂₂₂ construct.

Figure for reviewer. **35 μM NRD1 147-222 (cell) + 317 μM NAB3 191-261 (syringe).** 20 mM Kpi pH 7.0, 150 mM NaCl, 1 mM β -mercaptoEtOH, 25 °C, 19 inj x 2 μL

Supplementary Figure 4:

I believe that it would be more informative to show the HSQC overlays in one single figure with the chimera in grey, bound Nrd1 in blue and bound Nab3 in red. That will show better which peaks belong to each protein and which peaks are different.

It would also be good to have a plot of the chemical shift difference as a function of the amino acid sequence, to highlight regions similar or different.

We have changed the Supplementary Figure 4 following the reviewer's suggestions. The two heterodimer sub-spectra are overlaid with the spectra of the Nrd1-Nab3 chimera

The new panel B includes chemical shift difference plots comparing the peak positions of equivalent residues in the various chimeras and the heterodimer sub-spectra

Supplementary Table 1:

Could the authors add a line stating how many Noes were used between amino acids of Nrd1 and Nab3 (pseudo intermolecular Noes)?

We have added the requested information in the revised version of Supplementary Table 1

Results: The integrity of the Nrd1/Nab3 interface is crucial for cell survival. To validate the heterodimeric structure, the authors mutated hydrophobic residues of Nrd1. Why didn't the author mutate Gln205 and 217 that are described as involved in intermolecular hydrogen bonds and are conserved.

This is an excellent suggestion of the reviewer and indeed we considered to assay mutants at those positions. However, we opted for mutating hydrophobic residues because we thought that the potential results obtained for the Gln205/Gln217 mutants might be more difficult to interpret. Their replacement by hydrophobic residues in buried locations might induce a favorable effect that could compensate the energy penalty of the hydrogen bond loss. On the contrary, replacing them by polar groups might cause structural rearrangements of the hydrogen bond network to accommodate the mutation. In summary, we chose the mutations that we thought would be more informative for *in vivo* validation of the structure.

Reviewer #2 (Comments to the Authors (Required)):

Perez-Canadillas and colleagues report a structural analysis of the dimerization interaction between yeast RNA binding proteins Nab3-Nrd1. They use a number of biophysical techniques, predominantly NMR, to assess the flexibility and extent of order induced upon dimerization of the isolated Nrd1-binding domain of Nab3 and Nab3-binding domain of Nrd1. They conclude that the interaction interfaces results from packed helices and induced structure. The authors suggest that the structural arrangement of these two proteins is analogous to RNA14/15 assembly seen for the other major termination mechanism in budding yeast.

This study is a straightforward analysis that continues the probing of the solution structure of Nab3 and Nrd1 by these authors and the Stefl lab. The data is of high quality and the interpretations are reasonable. The main criticism is of the extent to which much new insight is offered in the report. The biological significance of some of the findings: the possible 2:1 stoichiometry between the proteins, the suggestion that the CID "destabilizes" the interaction, the use of the fused Nrd1-Nab3 domains, and the internally deleted Nrd peptide, is questionable. The Discussion is interesting but does not really result from the investigation reported here which is of more limited scope. In sum, the paper is satisfactory but incremental.

We would like to thank the reviewer for his/her critical comments. We honestly think that our work makes an important contribution to the current structural knowledge of the Nrd1/Nab3/Sen1 system. Although this system has been intensively studied at structural level, the structural bases of Nrd1/Nab3 heterodimerization have remained as a missing piece of the NNS pathway puzzle.

To avoid overinterpretations, we have been very cautious discussing some aspects of the work like the apparent 2:1 stoichiometry and the destabilizing effect of the CID. For example, we have stated that experimental errors in the determination of Nab3 and Nrd1 concentrations by UV could affect the stoichiometry value (N) obtained by ITC and that the Nrd1 NAID oligomerization could affect it as well. On the other hand, the difference between Nrd1 CID-NAID and Nrd1 NAID-RBD dissociation constants is larger than the experimental error, but still lower than one order of magnitude. Although the results are indicative of a possible destabilizing effect of the CID, it would be necessary to validate it in further studies.

The use of the fused Nrd1-Nab3 domains was a necessary approach to obtain structural information. We attempted the structural determination of the Nrd1/Nab3 heterodimer by NMR (using ¹³C/¹⁵N filtered and edited NOESY spectra) but the obtained data were low-quality. We also tried, unsuccessfully, to crystallize the heterodimer and even the Nrd1-Nab3 chimera. Maybe other researchers working on the structure of Nrd1 and Nab3 have faced similar technical problems, and we think that our approach has overcome them.

The deletion of the peptide between Nrd1 NAID and RBD domains was mandatory because the construct including the RE-rich sequence was insoluble.

Finally, the Discussion is based on the prediction (in light of our structure) of the Nrd1/Nab3 heterodimer being exclusive of the Saccharomices clade and, as an example, we speculate that this interaction is behind the differential functionality between *S. cerevisiae* Nrd1/Nab3 and their orthologs in *S. pombe*.

Minor comments:

- Number the pages please
- grammatical proofreading is needed
- what is meant by the Nab3 NRID "optimized" construct? Optimized for what?
- Define the acronyms in the Introduction: NMR, CD, ITC
- Typo: in results KD not KDs
- Typo in Supp Fig legend 3: ITC not ICT

All these minor comments have been included in the revised version of the manuscript. The term optimized has been removed from the text as it only referred to the trimming of the flexible N- and C-terminal ends of the Nab3 construct.

Careful proofreading of the text has been made and the found typos and grammatical errors were corrected.

Reviewer #3 (Comments to the Authors (Required)):

This is a very nice and simple paper that solves the structure of the Nrd1/Nab3 heterodimerization domains. Starting with a basic analysis of the individual proteins, the authors show that the Nab3 domain forms a small domain consisting of a few helices, and that Nrd1 folds around the Nab3 fold. I'm not an NMR specialist, but the structure looks very plausible and convincing. To solve a technical problem with the proteins, they used a clever trick of fusing the two domains together, but the spectra and other data argue that their proposed structure exists in the context of the full length proteins. In particular, *in vivo* complementation experiments show that mutations predicted to disrupt the dimer interface lead to growth defects. The discussion points out an interesting structural similarity of the Nrd1/Nab3 dimerization structure to that of the polyadenylation factors RNA14/RNA15, although this clearly reflects convergent evolution rather than a shared ancestor. Overall, I have only two small comments related to references:

We are grateful with the reviewer's comments about our work. We are particularly pleased that she/he finds interesting the discussion about the structural similarities between the two transcription termination pathways. Although we also agree with her/his judgement on the convergent evolution of RNA14/RNA15, it would be interesting to further investigate if the NNS pathway arose from a "copy" of the original poly(A) dependent one. In this sense, the whole genome duplication (WGD) experienced by *Saccharomyces cerevisiae* gave this organism the opportunity to develop new functions. Because we are not experts on evolutionary genetics, we cannot further explore this hypothesis, but hope that our work encourages other research groups to do it.

1. In the first paragraph of the introduction: I don't think the 2011 Marquardt paper is the right reference for showing that there are two pathways for termination. This paper is really more about degradation pathways. A better reference would be Kim *et al.* (2006) *Mol Cell* 24:723. Before this paper, it was less clear that the pA pathway and NNS pathway were really separate, rather than variations of the same mechanism.

The reference Marquardt *et al.* (2011) has been replaced by Kim *et al.* (2006) in the revised version of the manuscript.

2. In the discussion, I think it's overstating things to state that Tyr1 phosphorylation acts to block termination in the middle of genes. That was the model proposed in the Mayer 2012 paper based on circumstantial evidence (ChIP of Tyr1-P and peptide binding). However, mass spec experiments on the CTD find so little tyrosine phosphorylation on the CTD that it can't possibly be enough to block Ser5P and Ser2P binding proteins. I would just take out that sentence, as it's not necessary for the story presented here.

We have removed the mentioned sentence following the reviewer comment.

December 9, 2021

RE: Life Science Alliance Manuscript #LSA-2021-01252-TR

Dr. José Manuel Perez-Cañadillas
Institute of Physical-Chemistry
Biological Physical Chemistry
C/Serrano 119
MADRID, MADRID 28006
Spain

Dear Dr. Perez-Cañadillas,

Thank you for submitting your revised manuscript entitled "Structural basis of Nrd1-Nab3 heterodimerization". We would be happy to publish your paper in Life Science Alliance pending final revisions necessary to meet our formatting guidelines.

- supplementary references should be a part of the main references
- Please upload all figure files as individual ones, including the supplementary figure files; all figure legends should only appear in the main manuscript file
- please add the Twitter handle of your host institute/organization as well as your own or/and one of the authors in our system
- please be sure that all Authors have been added to the Author Contribution section in the manuscript text
- please add your main and supplementary figure legends to the main manuscript text after the references section
- please add callouts for Figures S1A and B; S3A and B; and S5A and B to your main manuscript text;

A. FINAL FILES:

B. MANUSCRIPT ORGANIZATION AND FORMATTING:

Sincerely,

Reviewer #1 (Comments to the Authors (Required)):

The authors have answered all my concerns. I have no further comments.

Reviewer #2 (Comments to the Authors (Required)):

The authors have responded admirably to many of the comments. The biggest criticism from this reviewer was the magnitude of the impact and general appreciation of the findings. The paper was well received from the other reviewers, providing assurance regarding the significance of these findings.

Reviewer #3 (Comments to the Authors (Required)):

I had only some minor comments that the authors addressed. I think they have also adequately addressed the critiques from the other reviewers. I therefore support publishing the current version of the paper.

December 15, 2021

RE: Life Science Alliance Manuscript #LSA-2021-01252-TRR

Dr. José Manuel Perez-Cañadillas
Institute of Physical-Chemistry
Biological Physical Chemistry
C/Serrano 119
MADRID, MADRID 28006
Spain

Dear Dr. Perez-Cañadillas,

Thank you for submitting your Research Article entitled "Structural basis of Nrd1-Nab3 heterodimerization". It is a pleasure to let you know that your manuscript is now accepted for publication in Life Science Alliance. Congratulations on this interesting work.

DISTRIBUTION OF MATERIALS:

Again, congratulations on a very nice paper. I hope you found the review process to be constructive and are pleased with how the manuscript was handled editorially. We look forward to future exciting submissions from your lab.

Sincerely,
